# Comparative Chloroplast Genomes Analysis Provided Adaptive Evolution Insights in *Medicago ruthenica*

**DOI:** 10.3390/ijms25168689

**Published:** 2024-08-09

**Authors:** Tianxiang Zhang, Manman Li, Xiaoyue Zhu, Shuaixian Li, Meiyan Guo, Changhong Guo, Yongjun Shu

**Affiliations:** Key Laboratory of Molecular Cytogenetics and Genetic Breeding of Heilongjiang Province, College of Life Science and Technology, Harbin Normal University, Harbin 150025, China; hsdztx@stu.hrbnu.edu.cn (T.Z.); limanman1208@stu.hrbnu.edu.cn (M.L.); zhuxiaoyue2001@126.com (X.Z.); hsdlsx1201@stu.hrbnu.edu.cn (S.L.); guomeiyan@163.com (M.G.); kaku3008@hrbnu.edu.cn (C.G.)

**Keywords:** *Medicago ruthenica*, chloroplast genomes, varieties, adaptive evolution

## Abstract

A perennial leguminous forage, *Medicago ruthenica* has outstanding tolerance to abiotic stresses. The genome of *Medicago ruthenica* is large and has a complex genetic background, making it challenging to accurately determine genetic information. However, the chloroplast genome is widely used for researching issues related to evolution, genetic diversity, and other studies. To better understand its chloroplast characteristics and adaptive evolution, chloroplast genomes of 61 *Medicago ruthenica* were assembled (including 16 cultivated *Medicago ruthenica* germplasm and 45 wild *Medicago ruthenica* germplasm). These were used to construct the pan-chloroplast genome of *Medicago ruthenica*, and the chloroplast genomes of cultivated and wild *Medicago ruthenica* were compared and analyzed. Phylogenetic and haplotype analyses revealed two main clades of 61 *Medicago ruthenica* germplasm chloroplast genomes, distributed in eastern and western regions. Meanwhile, based on chloroplast variation information, 61 *Medicago ruthenica* germplasm can be divided into three genetic groups. Unlike the phylogenetic tree constructed from the chloroplast genome, a new intermediate group has been identified, mainly consisting of samples from the eastern region of Inner Mongolia, Shanxi Province, and Hebei Province. Transcriptomic analysis showed that 29 genes were upregulated and three genes were downregulated. The analysis of these genes mainly focuses on enhancing plant resilience and adapting adversity by stabilizing the photosystem structure and promoting protein synthesis. Additionally, in the analysis of adaptive evolution, the *accD*, *clpP* and *ycf1* genes showed higher average Ka/Ks ratios and exhibited significant nucleotide diversity, indicating that these genes are strongly positively selected. The editing efficiency of the *ycf1* and *clpP* genes significantly increases under abiotic stress, which may positively contribute to plant adaptation to the environment. In conclusion, the construction and comparative analysis of the complete chloroplast genomes of 61 *Medicago ruthenica* germplasm from different regions not only revealed new insights into the genetic variation and phylogenetic relationships of *Medicago ruthenica* germplasm, but also highlighted the importance of chloroplast transcriptome analysis in elucidating the model of chloroplast responses to abiotic stress. These provide valuable information for further research on the adaptive evolution of *Medicago ruthenica*.

## 1. Introduction

*Medicago ruthenica* (2n = 2x = 16) is a close relative of the most important leguminous forage alfalfa (*Medicago sativa*), originating from grasslands and hilly areas in regions such as Siberia, Mongolia, and northern China [1]. As a perennial leguminous forage, it possesses superior characteristics such as cold tolerance, drought tolerance, and salt and alkali tolerance [2,3]. However, the stems of *Medicago ruthenica* do not stand upright, and its yield of forage and seeds is lower than that of alfalfa, which limits the widespread application of *Medicago ruthenica* [4,5]. It is believed to have higher soil nutrient use efficiency than cultivated alfalfa, making it potentially more suitable for low-input systems [6,7]. Reportedly, *Medicago ruthenica* exhibits better drought tolerance than alfalfa and other leguminous forage crops, and the hybrid varieties resulting from the crossbreed between *Medicago ruthenica* and *Medicago sativa* exhibit stronger tolerance to salinity-alkalinity and cold conditions compared to alfalfa [5,8]. This suggests its potential in improving the tolerance of *Medicago sativa* for growth under harsh environments where *Medicago sativa* cannot normally grow, therefore wild germplasm resources are widely used in modern *Medicago sativa* cultivation. The establishment of the high-quality genome of the model plant *Medicago truncatula* drove extensive resequencing of the *Medicago ruthenica* genome. Wang et al. and Yin et al. have constructed the genome of *Medicago ruthenica*, using these data to analyze its evolution and demographic history [7,9]. The chloroplast genome, which is an important component of genetic information, is lacking, with only two chloroplast genomes of *Medicago ruthenica* reported to date [10,11]. Studies on the structural differences between wild and cultivated genomes are widely used to reveal environmental adaptation and domestication processes [11,12]. Research on the chloroplast genome of *Medicago ruthenica*, both wild and cultivated varieties, can deepen our understanding of evolutionary mechanisms.

Chloroplasts in plant cells are believed to have originated from cyanobacteria subsumed by eukaryotic ancestors in at endosymbiosis events, gradually losing autonomy through integration with the host cell nucleus [13,14]. Chloroplasts are the crucial metabolic centers in the nucleus of plant cells, converting light energy into carbohydrates through photosynthesis [15,16]. In addition to participating in photosynthesis, chloroplasts also play a role in biological processes such as plant immunity and growth and development [17,18]. Chloroplast genomes exhibit a typical quadripartite circular structure, with highly conserved gene arrangements, ranging from 115 to 180 kb in length, divided into a large single copy (LSC) and a small single copy (SSC) region by two inverted repeats (IR) sequences [19,20]. In nature, there are some plants that lack an IR region, which are called inverted repeat-lacking clades (IRLC) [21]. It was found that the IR region was obviously lacking in the chloroplast of *Medicago ruthenica*, which was also observed in *Medicago sativa* [22].

The genetic information of chloroplasts is maternally inherited, resulting in a more conserved chloroplast gene structure. The highly conserved genome structure of chloroplast genomes ensures high assembly accuracy, making them useful for studies in phylogenetics, biogeography, and population genetics [12,23]. Qihang Chen et al. resolved the evolutionary and domestication history of East Asian peonies through the plastid genome [24]. The chloroplast genome contains important genetic information and is also the site of synthesis for many macromolecules, including amino acids and fatty acids [25]. Abiotic stress is a major factor limiting plant growth and development, which generally leads to a decrease in overall plant yield. Chloroplasts are highly sensitive to stress and generate reactive oxygen species (ROS) in response, which can overwhelm the antioxidant defense systems and cause irreversible damage to plants. Therefore, the chloroplast response to abiotic stress typically involves extensive proteomic changes. These changes at the gene level promote gene expression and regulation. For example, the Ndh complex of the chloroplast optimizes the induction of photosynthesis under water stress [26]. Additionally, abnormal chloroplast development can lead to incomplete grain development, resulting in reduced yield and quality [27]. Most chloroplast genes are functional and regulated, and are involved in various stress response mechanisms. Exploring the changes in transcription levels of chloroplasts induced by abiotic stress is of profound significance, not only because the complete chloroplast transcriptome has not been comprehensively reported in many plants, but also because such information will provide a potential mechanistic understanding at the transcriptome level.

In this study, 61 complete chloroplast genomes were by assembled integrating published genomic data of cultivated and wild *Medicago ruthenica* from eastern and western regions of China. The analysis focuses on phylogenetic relationships and genetic diversity, especially the divergence of chloroplast populations in different regions, which provides new chloroplast genomic evidence for the origin and evolution of *Medicago ruthenica*. Meanwhile, chloroplast genes involved in responding to abiotic stress were identified by comparative transcriptomics, and their roles in mediating tolerance in *Medicago ruthenica* were discussed.

## 2. Results

### 2.1. General Features of Medicago ruthenic Chloroplast Genomes

This study successfully assembled 61 chloroplast genomes using resequencing data (Appendix A). All chloroplast genomes belong to the IRLC and lack IR regions, which is consistent with the two previously published *Medicago ruthenica* chloroplast genomes. The total length of all *Medicago ruthenica* chloroplast genomes are 7,781,933 bp, ranging from 127,233 bp (W7, collected from Gannan) to 127,998 bp (W1, collected from Jiuquan). The guanine–cytosine (GC) content varies slightly, ranging from 34.16% (E27, collected from Beian) to 34.22% (W14, collected from Tongren) (Appendix A). A total of 108 unique genes were annotated in *Medicago ruthenica* chloroplast genomes, including 74 protein-coding genes, 30 tRNA genes, and four rRNA genes (Table 1, Figure 1). Among the protein-coding genes, 42 genes are associated with photosynthesis, 26 genes are related to self-replication, four other genes (*matK*, *clpP*, *cemA*, *accD*, *ccsA*), and four genes with unknown functions. The gene count is the same for all 61 chloroplast genomes and matches with all previously published *Medicago ruthenica* chloroplast genomes to date. Among the 74 protein-coding genes, 10 genes contain one intron (*ndhA*, *ndhB*, *petB*, *petD*, *atpF*, *rpl16*, *rpl2*, *rps12*, *rpoC1*, *clpP*) and one gene contains two introns (*ycf3*) in the 61 chloroplast genomes. Additionally, there are five tRNA genes (*trnA-UGC*, *trnI-GAU*, *trnK-UUU*, *trnL-UAA*, *trnV-UAC*) that contain one intron.

### 2.2. Comparative Chloroplast Genome Analysis

A pan-chloroplast genome was constructed based on 61 *Medicago ruthenica* complete chloroplast genomes, using the pan-chloroplast genome as a reference, and a comparative analysis was conducted using the mVISTA software (https://genome.lbl.gov/vista/index.shtml, accessed on 18 November 2023). The visualization results indicated that coding regions are more conserved than non-coding regions (Figure 2). By comparing cultivated and wild varieties from eastern and western regions, relatively high levels of variation in the *accD* and *clpP* were found in various regions. Most of the protein-coding genes were highly conserved. It was observed that the *petB (exon2)-petB (exon1)*, *accD-trnQ*, *psbZ-trnS*, and *rrn16S-trnI-exon1* regions showed high variability in the eastern region, while in the western region, *petB (exon2)-petB (exon1)*, *petN-rpl20*, *psbZ-trnS*, *and rrn16S-trnI-exon1* regions exhibited relatively high variability. These highly variable regions in the five genomes were mainly located in intergenic regions.

### 2.3. Adaptive Evolution of the Medicago ruthenica Chloroplast Genomes

Using the pan-chloroplast genome to evaluate the chloroplast genomes of 61 *Medicago ruthenica*, analyzed nucleotide variability with a conserved length of 125,064 bp (Figure 3, Appendix A). Compared to coding regions, intergenic regions showed higher nucleotide diversity, with significantly higher Pi values (>0.005) found in *petB (exon2)-petB (exon1)* and *petB (exon2)*. Among them, the nucleotide variability in the *accD* variable region was the highest, with a Pi value of 0.00571. Analysis of Ka/Ks ratio of protein-coding genes from 61 chloroplast genomes was conducted to explore adaptive evolution (Figure 4, Appendix A). The Ka/Ks ratio of all protein-coding genes ranged from 0 to 4.216, with most of them exhibiting a relatively low average Ka/Ks ratio, indicating conservatism in the evolutionary process of chloroplast genes of *Medicago ruthenica*. Only four protein-coding were are under positive selection, with *ycf1* having the highest average Ka/Ks ratio among them.

### 2.4. Phylogenetic Analyses and Haplotype of Medicago ruthenica

The phylogenetic tree was constructed using the whole complete chloroplast genome sequences with the ML method (Figure 5a). In the phylogenetic tree, *Medicago ruthenica* were clustered into two major clades based on the complete chloroplast genomes. One clade comprised all the cultivated and wild varieties from the eastern regions, while the other clade contained the remaining varieties from the western regions. Based on the results of chloroplast genome alignment, we constructed a median-joining network, where 61 *Medicago ruthenica* samples were divided into 56 haplotypes (Figure 5b, Table 2). The varieties from the eastern region included 27 haplotypes, and the varieties from the western region included 29 haplotypes, indicating rich haplotype diversity. Hap_45 was identified as a common haplotype shared by four varieties from the western region (Gansu and Qinghai Provinces). In the haplotype analysis, it was found that Hap_13 and Hap_14 belong to the haplotypes of the eastern region, but they are specific to W1 (Jiuquan, Gansu) and W3 (Gulang, Gansu). Correspondingly, in the phylogenetic tree, W1 and W3 were also excluded from the classification of the western region. This might be due to the extensive hybridization and gene flow between different varieties of *Medicago ruthenica*, resulting in a mixed and diverse chloroplast genome.

### 2.5. Phylogenetic Tree Based on SNPs

The 61 complete chloroplast genomes were compared with the chloroplast genome of Taihang, revealing 785 SNP variant sites and constructing an ML phylogenetic tree based on the constant sites (Figure 6). Interestingly, the phylogenetic results based on variant sites differed from the abovementioned results based on the complete chloroplast genomes. The 61 chloroplast genomes were divided into three groups. Analysis of the geographic distribution of the varieties in each clade indicated that not all individuals from the same region were grouped into the same genetic clade. Varieties from Gansu, Qinghai, and Sichuan Provinces form one cluster of western regions. Varieties from the eastern region of Inner Mongolia, Heilongjiang, and Jilin are grouped into the same genetic cluster. Then, the remaining eastern region of Inner Mongolia, with Xian and Hebei Province as the main varieties, forms a branch representing the intermediate region.

### 2.6. Response of Chloroplast Genes to Abiotic Stress in Medicago ruthenica

Transcriptomic analysis of the chloroplast genome in *Medicago ruthenica* under various types of stress reveals significant gene expression changes (Figure 7a, Appendix A). Under non-biological stress conditions, three genes were significantly downregulated (Figure 7c, Appendix A), while 29 genes were upregulated (Figure 7b, Appendix A). Overall, chloroplasts exhibit negative regulation of photosynthesis-related genes and positive responses in most tRNA genes under abiotic stress. In comparison, low-temperature stress induces more extensive gene expression changes, predominantly upregulation, indicating a complex and adaptive response of plants to cooling environmental changes. Additionally, ABA (abscisic acid), as a crucial plant hormone, promotes the responses of the chloroplast genome of *Medicago ruthenica* to abiotic stress.

### 2.7. Analysis of RNA Editing Efficiency

By combining transcriptome data, 21 editing events were identified in *Medicago ruthenica* under control (Figure 8, Appendix A). Under abiotic stress, 34 editing events were identified through RNA-Seq mapping, with a potential of 214 RNA editing sites. Among these, C-to-A and C-to-T were the main editing events, with *ycf1* and *clpP* having the most editing events. Compared to the control, the RNA editing efficiency of *psaB*, *atpA*, and *atpH* significantly increased under abiotic stress. This result suggests that RNA editing may be related to the synthesis and function of protease subunits and ATP synthase subunits.

## 3. Discussion

*Medicago ruthenica* is a perennial leguminous forage with abundant genetic diversity and super tolerance to abiotic stress [3]. This provides valuable genetic resources for improving traits related to abiotic stress tolerance in legume forage. Species identification and phylogenetic studies based on the chloroplast genome have received increasing attention from researchers [28]. So far, there have been only two reported studies on the chloroplast genome of *Medicago ruthenica*, and there has yet to be any comparative research on the chloroplast genome of *Medicago ruthenica* [10,11]. In our study, we obtained 61 new complete circular chloroplast genomes, with an average length of 127,572 bp, from wild and cultivated *Medicago ruthenica* varieties using the GetOrganelle toolkit [29]. The pan-chloroplast genome was constructed from these chloroplast genomes, and differs from the typical chloroplast genome structure [20]. Although the chloroplast genome of alfalfa exhibits a highly conserved structure, it contains only one LSC region, one SSC region, and one IR region. Recombination has occurred multiple times in the chloroplast genomes of *Medicago ruthenica* during evolutionary and developmental processes, resulting in the complete loss of an inverted repeat sequence, as observed in *Medicago truncatula* and *Cicer arietinum* [30,31]. Compared to the ancestral angiosperm chloroplast genome, seven coding genes are missing [32]. Notably, *infA* is also completely absent, including in *Medicago sativa* and *Cicer arietinum* [30].

Despite the conserved structure and genes of the chloroplast genome, a large amount of nucleotide variation was found in the chloroplast of *Medicago ruthenica*. These nucleotide variations can be used to develop molecular markers for the identification of *Medicago ruthenica* and to distinguish among different varieties [33]. Among all identified CDS variations, it was found that the number of mutations in the *accD* of the *Medicago ruthenica* chloroplast genome far exceeded that of other genes. The variation hotspot regions in the chloroplast genome can be used for studying evolution, DNA barcoding, and accurate, effective molecular markers for a taxon [33,34]. In previous studies, Kode and Katayama found that *accD* could serve as a potential chloroplast molecular marker for phylogenetic and genetic diversity analysis in *Nicotiana tabacum* and *Pyrus* spp. [35,36]. Further analyzing the variations in *accD*, we found that the diversity of *accD* was richer in the eastern regions than the western regions, providing the possibility to distinguish between eastern and western varieties of *Medicago ruthenica*. In addition to *accD*, more informative nucleotide variations were also found in the *clpP* in study varieties. This gene has been studied for adaptive evolution in the tribe *Sileneae* and *Oenothera* [37]. The number of insertions and deletions found in CDS was relatively low, with only two genes containing highly polymorphic SNP loci.

The number of insertions and deletions found in non-coding regions was higher than in other regions, which is consistent with studies related to evolutionary rates of coding and non-coding regions under selection [34]. Five highly variable regions were detected in the complete genome of *Medicago ruthenica*, namely (*petB (exon2)-petB (exon1)*, *accD-trnQ*, *psbZ-trnS*, *rrn16S-trnI (exon1)*, *petN-rpl20)*, which are suitable for species identification. Similarly, *psbZ-trnS* has been reported as a potential molecular marker in *Ilex dabieshanensis*, while *rbcL-accD* is used for DNA barcoding in *Ligustrum lucidum* [38,39]. The coding region *accD* exhibits a relatively fast evolutionary rate across different species, and this gene is commonly used as a chloroplast DNA barcode. Additionally, *accD-trnQ* shows high variability among alfalfa populations from different regions, indicating that its non-coding sequences may further facilitate the phylogenetic analysis and variety identification of *Medicago ruthenica* from different geographical areas.

The Ka/Ks ratio is widely used to measure selective pressure and is associated with gene adaptive evolution [40]. Compared to non-synonymous (KS) substitutions, synonymous (KA) nucleotide substitutions are more frequent in most genes of organisms, so the Ka/Ks value is typically less than 1 [40,41]. A Ka/Ks ratio greater than 1 indicates positive selection, while a Ka/Ks ratio less than 1 indicates purifying selection [42]. In this study, the Ka/Ks ratio of most protein-coding genes was less than 1, indicating that they were under purifying selection. Additionally, the detection results indicate that four genes are strongly positively selected (*accD*, *clpP*, *ycf1*, *ycf2*). Non-synonymous substitutions can induce changes in amino acids, driving alterations in gene function to adapt to environmental pressures [28]. The results of this study showed that four genes (*accD*, *clpP*, *ycf1*, *ycf2*) were under positive selection. The analysis results of KS and KA rates showed that the *ycf1* had the highest average Ka/Ks ratio in the chloroplast genome of *Medicago ruthenica*. Similar to most plant studies, *ycf1* and *ycf2* are among the longest genes in the chloroplast genome, present in almost all plant chloroplasts, and can also identify different varieties in orchids and cucumbers [43,44]. *ClpP* encodes the caseinolytic protease complex, which plays an essential role in maintaining protein homeostasis and includes both plastid-encoded and nuclear-encoded subunits [45]. The *accD* encodes the β-carboxylase enzyme, which is a key enzyme in the biosynthesis of fatty acids. The *matK*, *ycf1*, *accD*, *rps3*, and *rpoA* were observed to positive selection in Juglans [46]. Therefore, the positive selection observed in these genes across different varieties indicates that they are undergoing rapid evolution [47]. Further research into their functions could be crucial for understanding the adaptive evolution and breeding of *Medicago ruthenica*.

With the rapid development and improvement of sequencing technologies and analytical approaches, an increasing number of chloroplast genome data have been successfully utilized in phylogenetic studies [28,33,34]. Using 61 chloroplast genomes, we reconstructed a phylogenetic tree within *Medicago ruthenica*, and results were supported by the genomic clustering analysis conducted by Wang et al. and Yin et al. [7,9]. In the classification of *Medicago ruthenica*, wild and cultivated varieties are clearly separated. E1 and E4, as well as E8 and other varieties, were divided into two separate branches, which are closely related to wild species. E6 and E9 were grouped into an independent branch, indicating a closer relationship between these two cultivated varieties compared to other wild cultivated varieties (Figure 5a and Figure 6). The classification of W1 and W3 varieties from Gansu Province together with those from the eastern region may be due to the self-incompatibility of *Medicago ruthenica*, which leads to extensive hybridization and gene flow among different varieties [48]. As a result, certain varieties may not be classified according to their geographical regions. The results indicate that studying species relationships based on complete chloroplast genome data is reliable at the intra-specific level. However, the phylogenetic relationships based on variant sites differed from those based on complete chloroplast genomes, grouping some eastern region of Inner Mongolia varieties with Hebei and Xian, forming a Central region. All alfalfa varieties are thus divided into three categories: eastern, central, and western. This is likely due to the eastern region’s chloroplast genomes containing more variability, as evidenced by the higher number of haplotypes observed in the eastern region.

RNA editing analysis reveals a significant increase in the RNA editing efficiency of *clpP* and *ycf1* in the chloroplast genome under abiotic stress conditions, which may represent an adaptive response mechanism of *Medicago ruthenica* to external stress. The increased RNA editing helps regulate the expression levels of chloroplast genes, thereby enhancing plant adaptation to abiotic stress [49]. *clpP* is a subunit of the chloroplast protease complex involved in protein folding and degradation processes. As a component of the chloroplast protease complex, *clpP* may contribute to stress adaptation by clearing abnormal proteins and maintaining protein homeostasis within the chloroplast [50,51]. The *ycf1* gene contains a long open reading frame, potentially involved in regulating the transcription and translation processes of chloroplast genes, thereby influencing the function of chloroplast proteins. Plants have evolved complex mechanisms to adapt to various environments [5,16].

Transcriptome analysis has revealed the upregulation of 29 genes in the chloroplast genome under abiotic stresses, including some encoding components of Photosystem II and I. Environmental stress significantly impacts the balance between high-energy electron generation in photosynthesis and Calvin–Benson–Bassham. Processes like energy transfer to Photosystem II and I reaction centers are activated to decrease photoinhibition and harmful effects from excessive reactive oxygen [52]. For instance, cold-induced translation increases PetL protein levels to regulate cold adaptation in tobacco [53]. These Photosystem II genes are speculated to enhance plant adaptation to adversity by stabilizing the Photosystem II structure. Overall, abiotic stress tends to inhibit plant photosynthesis, impacting chloroplast stability and inducing chloroplast degradation [54], leading to a reduction in genes associated with photosynthesis. However, tRNA, a critical factor influencing translation, plays a significant role in regulating gene expression in chloroplasts under abiotic stress, thereby contributing to the stability and efficiency of protein synthesis [55]. These findings not only deepen our understanding of how plants respond to environmental stress but also provide important clues for exploring adaptation mechanisms in *Medicago ruthenica*.

## 4. Materials and Methods

### 4.1. Assembly and Annotation of Chloroplast Genomes

The raw sequence data of 70 *Medicago ruthenica* leaf DNA sequencing datasets (19 cultivated species and 51 wild species) were received from the SRA database (https://www.ncbi.nlm.nih.gov/sra) (accessed on 20 December 2023) on the NCBI website including PRJNA598783 and PRJNA692663. The fastq files of DNA sequencing data were extracted from SRA files using the fastq-dump tool in the SRA toolkit (https://ftp-trace.ncbi.nlm.nih.gov/sra/sdk/2.9.6/, accessed on 4 January 2024). The raw reads were trimmed and quality controlled using Trimmomatic software v0.39 [56] to remove low-quality reads with a phred score less than 20 and a length less than 50. The filtered reads were de novo assembled using GetOrganelle software (version 1.7.5) [29] with *Medicago ruthenica* ‘Taihang’ (GenBank accession number: MW703984.1) as a reference and the k-mer lengths were set to 21, 45, 65, 85, and 105. The contigs were manually checked and adjusted for errors using Geneious Prime 2022 software [57] based on the start and stop codons of *Medicago ruthenica* ‘Taihang’. To demonstrate the accuracy of the assembly, the complete chloroplast genome was aligned with *Medicago ruthenica* ‘Taihang’ using MUMmer4.0 [58] for synteny analysis. Then, each assembled complete chloroplast genome was annotated and error-corrected using CPGAVAS2 software (version 2.0) and CPGview-RSG software (http://47.96.249.172:16019/analyzer/home, accessed on 27 January 2024) [59], with the reference genome *Medicago ruthenica* ‘Taihang’ and default parameters. Additionally, seqkit (version 2.5.0) was used to calculate sequence length and GC content [60]. The physical maps of pan-chloroplast genomes of *Medicago ruthenica* were drawn using online software Chloroplot (https://irscope.shinyapps.io/Chloroplot/, accessed on 29 January 2024) [61].

### 4.2. Divergence Analysis of Medicago ruthenica

Using the mVISTA online software (https://genome.lbl.gov/vista/index.shtml, accessed on 18 November 2023) [62], the sequence divergence among *Medicago ruthenica* species was visualized. The pan-chloroplast genome was used as reference in the shuffle-LAGAN model, where cultivated and wild varieties from the western and eastern regions were compared. The minimum and maximum Y-axes were set to 50% to 100%, and the sliding window size was set to 100 bp. To identify the sequence divergence of the complete chloroplast genome of *Medicago ruthenica*, we aligned the assembled chloroplast genome with the pan-chloroplast genome using MAFFT software (version 7.487) [63], with default parameters. Subsequently, nucleotide variability (Pi) values were analyzed using DnaSP 6 software (version 6.12.03) [64] to evaluate sequence divergence. The step size was set to 200 bp, and the window length was set to 1000 bp.

### 4.3. Phylogenetic Tree and Haplotype Analysis

The complete chloroplast genome of 61 *Medicago ruthenica* germplasm were aligned with the nucleotide sequence of the *Medicago ruthenica* ‘Taihang’ using MAFFT (version 7.487) [10,63] to construct a phylogenetic tree. An ML tree was reconstructed using IQtree 2 (version 2.0) [65], with 1000 bootstrap replicates for the alignment to build the phylogenetic tree, under the best model: UNREST + FO + R10. To visualize and modify the phylogenetic tree, FigTree software (version 1.4.4) was employed [66]. After the homologous sequence alignment, DnaSP 6 software (version 6.12.03) [64] was used to calculate the total number of haplotypes. Additionally, we used POPART (version 1.7) [67] to analyze the gene flow diversity of haplotypes and construct the haplotype network for all *Medicago ruthenica* species. Based on the sequence alignment results, the KaKs_Calculator 3.0 software [68] was used to calculate the nonsynonymous (Ka) and synonymous (Ks) substitution rates as well as the Ka/Ks ratio for all complete chloroplast genes of *Medicago ruthenica*.

### 4.4. Variants Calling

Filtered reads of 61 resequencing data were separately mapped to the ‘Taihang’ chloroplast genome using Burrows-Wheeler Aligner (BWA) software (version 0.7.17) [69]. Alignment files were converted SAM (Sequence Alignment Map) files into sorted BAM (binary version of SAM) files with SAMtools [70]. Next, the addOrReplaceReadGroups tool of GATK (version 4.4.0.0) was used for read group addition, followed by duplicate removal using the MarkDuplicates tool [71]. Then, we performed joint genotyping on gVCF files produced by HaplotypeCaller tool and converted the GVCF files into VCF files using the GenotypeGVCFs tool. Finally, SNP and indel variants were separated by SelectVariants.

### 4.5. Population Structure and Phylogenetic Tree Based on SNPs

SNPs were filtered using VCFtools (version 0.1.17) [72] with a missing rate lower than 50%, a minor allele count higher than 3, and a minor allele frequency higher than 0.05. Filtered SNPs were aligned using MAFFT (version 7.487) [63] with default parameters and concatenated to build a phylogenetic tree with IQTREE 2 (version 2.0) [65]. The best-fitting model determined by ModelFinder was F81 + F + R2. The tree construction involved 1000 replicates of ultrafast bootstrap and maximum likelihood. Visualization of the phylogenetic tree was performed using FigTree software (version 1.4.4) [66].

### 4.6. RNA-Seq Analysis

A total of six accessible RNA sequencing data of *Medicago ruthenica* were received from the SRA database (https://www.ncbi.nlm.nih.gov/sra, accessed on 2 March 2024) on the NCBI website, including SRR4140266. The clean reads from the six RNA-Seq libraries were mapped to *Medicago ruthenica* chloroplast using Salmon software 1.10.0 [73]. We calculated TPM values using default parameters and visualized them with an R package. Raw reads of six resequencing data were separately mapped to ‘Taihang’ using Burrows-Wheeler Aligner (BWA) software [69]. Alignment files converted SAM (Sequence Alignment Map) files into sorted BAM (binary version of SAM) files with SAMtools (version 1.19.2) [70]. Then, the GATK addOrReplaceReadGroups tool was used for read group addition, followed by duplicate removal using the MarkDuplicates tool [71]. Then, we performed joint genotyping on gVCF files produced by the HaplotypeCaller tool and converted the GVCF files into VCF files using the GenotypeGVCFs tool. SNP variants were separated by SelectVariants to obtain a VCF file containing only SNPs. REDO software (version 1.0) was used to detect potential RNA editing sites. Finally, we visualized the RNA editing sites using an R package [74].

## 5. Conclusions

In this study, a total of 61 chloroplast genomes from eastern and western regions of China were assembled, with sizes ranging from 127,233 bp to 127,998 bp. In addition, we identified highly polymorphic SNP sites located in *petB (exon2)-petB (exon1)* and the accD gene. These sites will be used as candidate SNP markers in future studies. In the phylogenetic tree and haplotypes, the chloroplast genomes of *Medicago ruthenica* have two major clades, distributed in the eastern and western regions. However, according to the chloroplast genome variation information, varieties from the central region form an independent clade. However, more varieties are still needed for further research to explore the origin and evolution of *Medicago ruthenica*. The *accD*, *clpP*, and *ycf1* genes exhibit higher average Ka/Ks ratios and significant nucleotide diversity, showing strong positive selection. Transcriptome results showed that under abiotic stress, 29 genes were upregulated, and three genes were downregulated. These genes are mainly involved in enhancing plant stress resistance and adaptation by stabilizing photosystem structures and promoting protein synthesis. Overall, our results not only enrich the complete chloroplast genome resources of *Medicago ruthenica* but also provide useful information on the adaptive evolution of chloroplast genes.

## Figures and Tables

**Figure 1 ijms-25-08689-f001:**
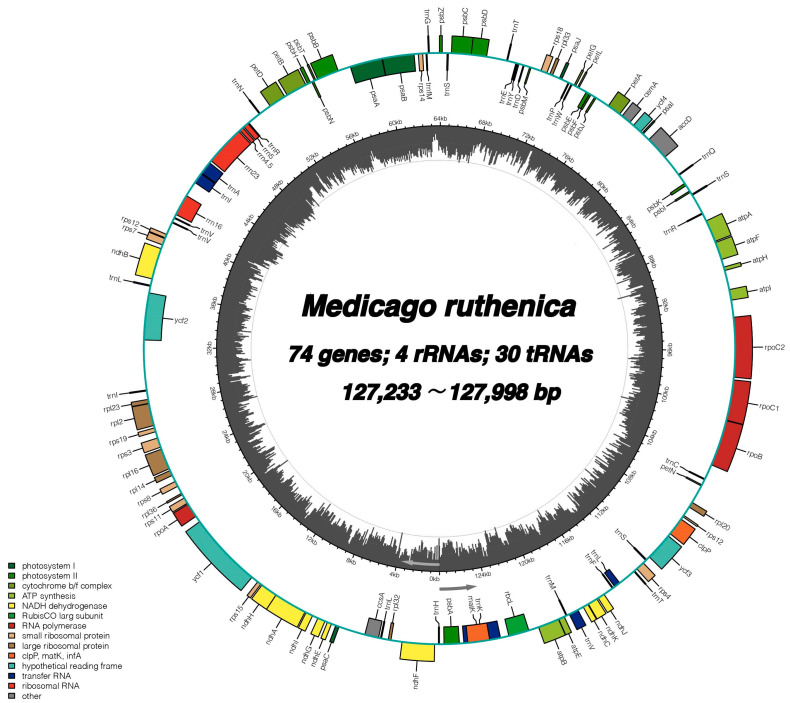
The pan-chloroplast of *Medicago ruthenica*. Genes shown on the outside of the first outer ring are transcribed counter-clockwise, while those on the inside are transcribed clockwise. Different functional groups of genes are color-coded. The dark grey bars on the second outer ring correspond to GC content.

**Figure 2 ijms-25-08689-f002:**
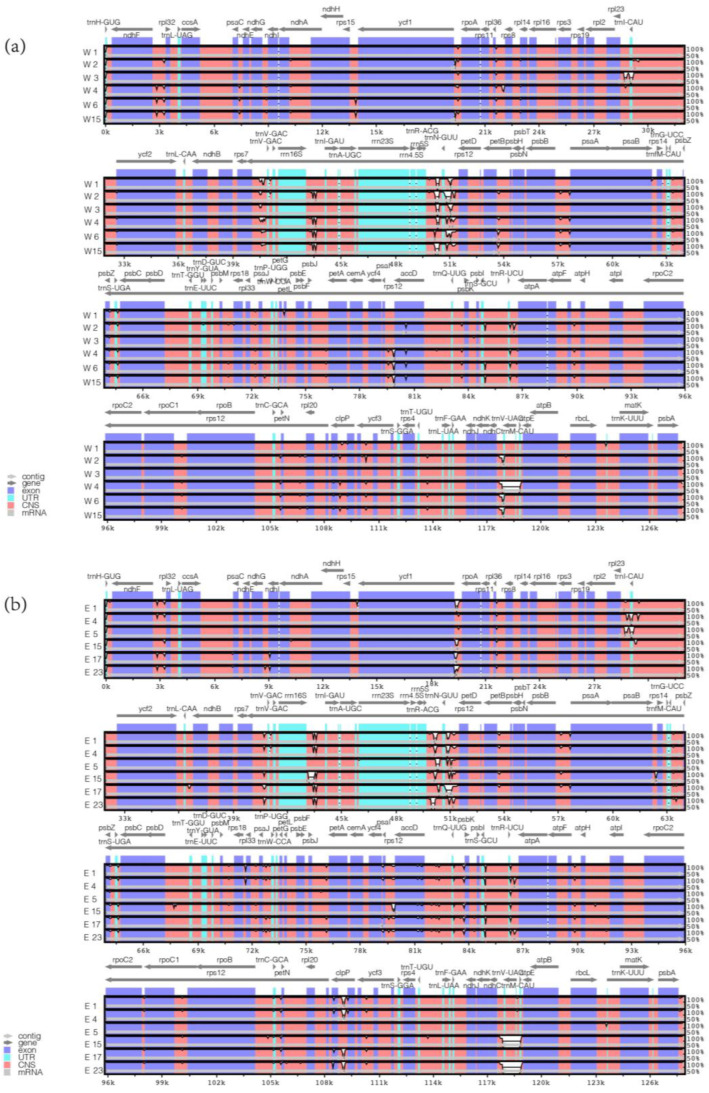
The complete chloroplast genome sequences of *Medicago ruthenica* were visualized with the pan-chloroplast genome as a reference. Gray arrows and thick black lines represent the orientation of genes. Purple bars indicate exons, sky-blue bars denote untranslated regions (UTRs), and red bars highlight non-coding sequences (CNS). Gray bars correspond to mRNA, while white regions show sequence differences among all analyzed chloroplast genomes. The horizontal axis shows the positions within the chloroplast genome, and the vertical scale indicates the identity percentage, ranging from 50% to 100%. (**a**) W1, W2, and W3 are cultivated varieties from the western region, while W4, W6, and W15 are wild varieties from the western region. (**b**) E1, E4, and E5 are cultivated varieties from the eastern region, while E15, E17, and E23 are wild varieties from the eastern region.

**Figure 3 ijms-25-08689-f003:**
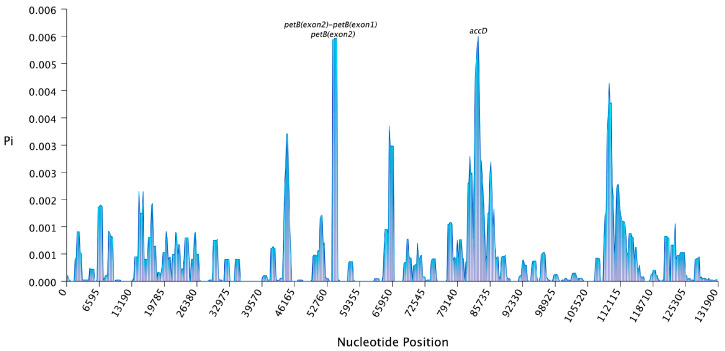
The nucleotide diversity (π) values in *Medicago ruthenica* chloroplast genome. Nucleotide diversity by sliding window analysis in 61 complete chloroplast genomes. Window length: 1000 bp, step size: 200 bp.

**Figure 4 ijms-25-08689-f004:**
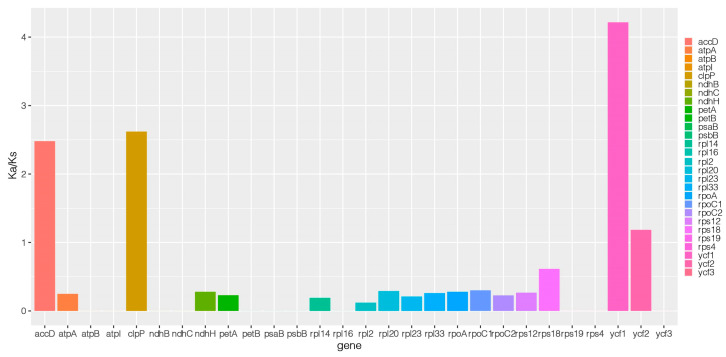
The evolutionary values of the Ka/Ks ratio in *Medicago ruthenica*.

**Figure 5 ijms-25-08689-f005:**
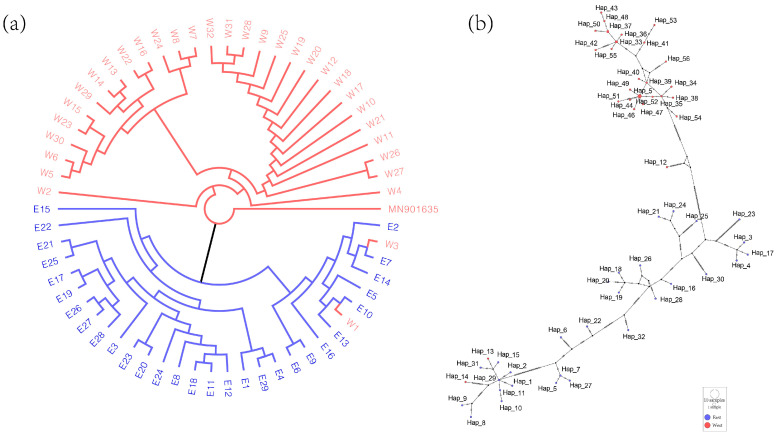
Phylogenetic relationships and haplotypes of *Medicago ruthenica* based on complete chloroplast genome sequences. (**a**) The phylogenetic tree was established by the maximum likelihood method, with bootstrap replications set to 1000. Red represents the varieties from the western region, and blue represents the varieties from the eastern region, corresponding to the color scheme throughout the figure. (**b**) Haplotype network of *Medicago ruthenica* chloroplasts. Each circle in the haplotype network represents a unique haplotype, with the size of the circle proportional to the frequency of the haplotype. Lines connecting the circles indicate mutational steps between haplotypes. The colors within the circles are consistent with those in the phylogenetic tree.

**Figure 6 ijms-25-08689-f006:**
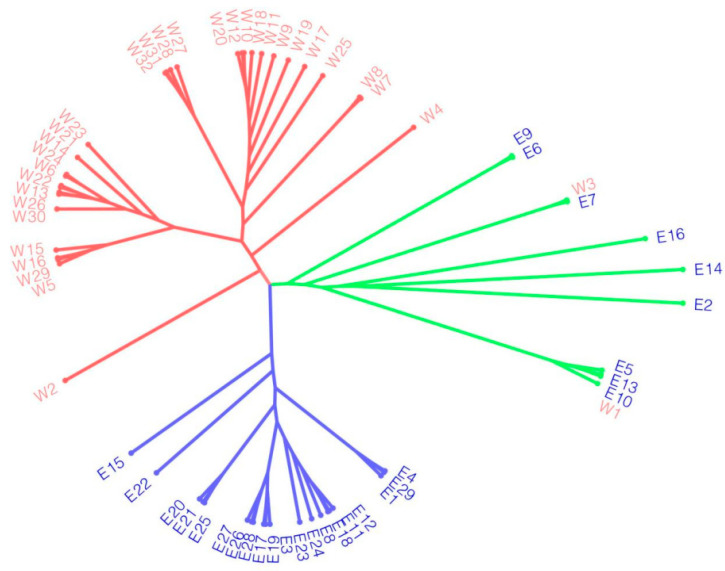
The phylogenetic tree was constructed based on SNPs across the genomes of the *Medicago ruthenica*. Red represents varieties from the western region, blue represents varieties from the eastern region, and green represents varieties from the intermediate region.

**Figure 7 ijms-25-08689-f007:**
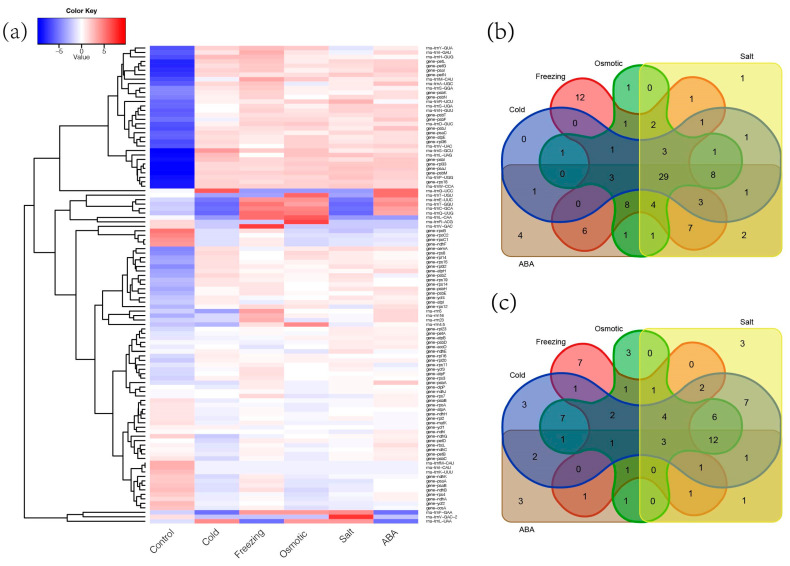
(**a**) Heatmap of chloroplast gene expression in *Medicago ruthenica* under abiotic stress. (**b**) Number of upregulated genes under different abiotic stresses. (**c**) Number of downregulated genes under different abiotic stresses.

**Figure 8 ijms-25-08689-f008:**
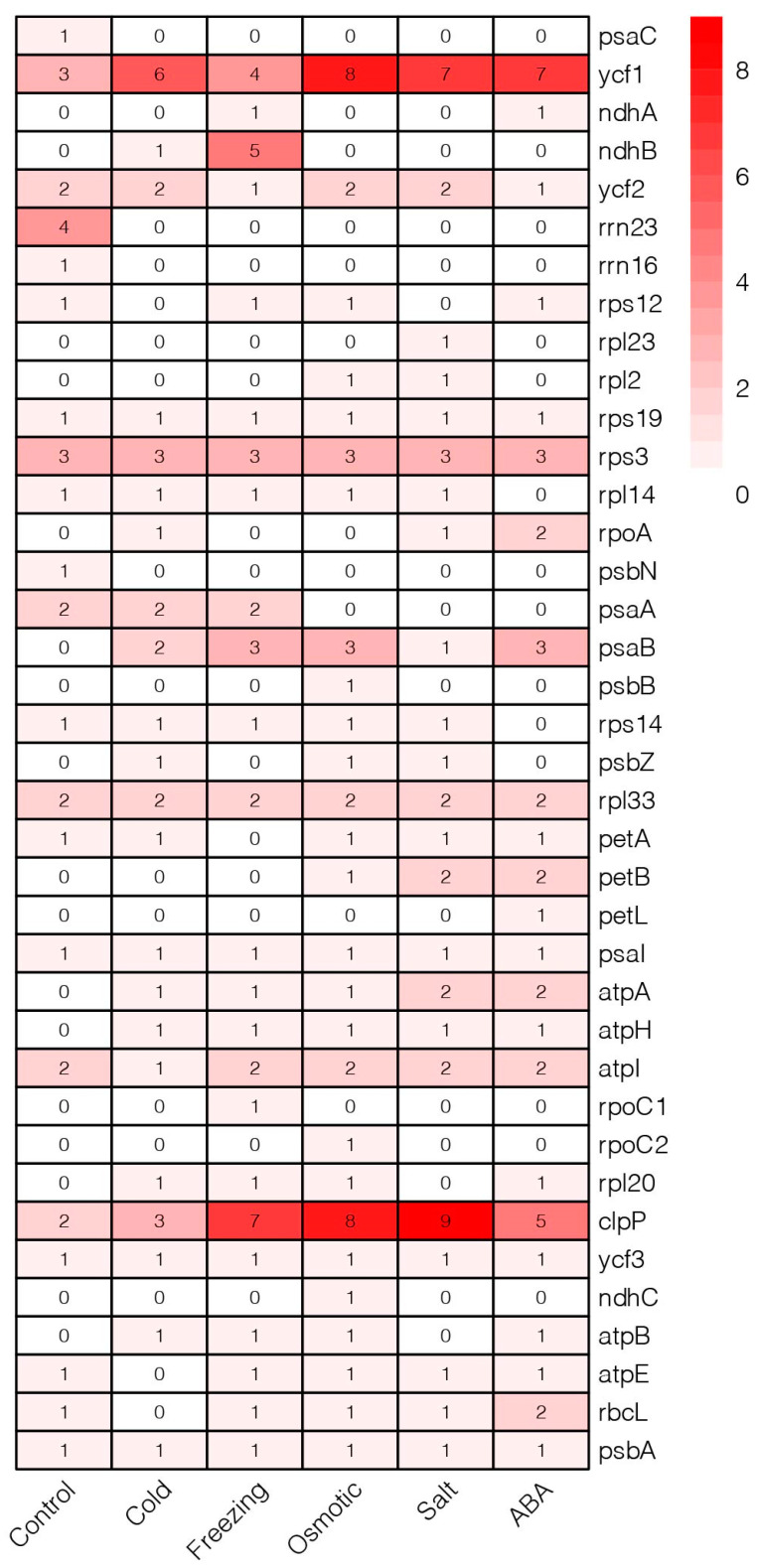
RNA editing efficiency in chloroplast genes under abiotic stress in *Medicago ruthenica*.

**Table 1 ijms-25-08689-t001:** Gene contents in 61 complete chloroplast genomes of *Medicago ruthenica*.

Gene Category	Functional Group	Gene Name
Photosynthesis	Subunits of photosystem I	*psaA*, *psaB*, *psaC*, *psaI*, *psaJ*
	Subunits of photosystem II	*psbA*, *psbB*, *psbC*, *psbD*, *psbE*, *psbF*, *psbH*, *psbI*, *psbJ*, *psbK*, *psbM*, *psbN*, *psbT*, *psbZ*
	Subunits of NADH dehydrogenase	*ndhA* *, *ndhB* *, *ndhC*, *ndhE*, *ndhF*, *ndhG*, *ndhH*, *ndhI*, *ndhJ*, *ndhK*
	Subunits of cytochrome b/f complex	*petA*, *petB* *, *petD* *, *petG*, *petL*, *petN*
	Subunits of ATP synthase	*atpA*, *atpB*, *atpE*, *atpF* *, *atpH*, *atpI*
	Large subunit of rubisco	*rbcL*
	Subunits photochlorophyllide reductase	*-*
Self-replication	Proteins of large ribosomal subunit	*rpl14*, *rpl16* *, *rpl2* *, *rpl20*, *rpl23*, *rpl32*, *rpl33*, *rpl36*
	Proteins of small ribosomal subunit	*rps11*, *rps12* *, *rps14*, *rps15*, *rps18*, *rps19*, *rps3*, *rps4*, *rps7*, *rps8*
	Subunits of RNA polymerase	*rpoA*, *rpoB*, *rpoC1* *, *rpoC2*
	Ribosomal RNAs	*rrn16S*, *rrn23S*, *rrn4.5S*, *rrn5S*
	Transfer RNAs	*trnA-UGC* *, *trnC-GCA*, *trnD-GUC*, *trnE-UUC*, *trnF-GAA*, *trnG-UCC*, *trnH-GUG*, *trnI-CAU*, *trnI-GAU* *, *trnK-UUU* *, *trnL-CAA*, *trnL-UAA* *, *trnL-UAG*, *trnM-CAU*, *trnN-GUU*, *trnP-UGG*, *trnQ-UUG*, *trnR-ACG*, *trnR-UCU*, *trnS-GCU*, *trnS-GGA*, *trnS-UGA*, *trnT-GGU*, *trnT-UGU*, *trnV-GAC(2)*, *trnV-UAC* *, *trnW-CCA*, *trnY-GUA*, *trnfM-CAU*
Other genes	Maturase	*matK*
	Protease	*clpP* *
	Envelope membrane protein	*cemA*
	Acetyl-CoA carboxylase	*accD*
	c-type cytochrome synthesis gene	*ccsA*
	Translation initiation factor	*-*
	other	*-*
Genes of unknown function	Conserved hypothetical chloroplast ORF	*ycf1*, *ycf2*, *ycf3* **, *ycf4*

Notes: *: gene containing one intron; **: gene containing two introns.

**Table 2 ijms-25-08689-t002:** Haplotype classification of 61 *Medicago ruthenica* chloroplast genomes.

Category	Number	Varieties
Hap_33	2	W5, W6
Hap_37	2	W14, W29
Hap_45	4	W10, W12, W19, W20

Note: Each of the other haplotypes corresponds to only one variety.

## Data Availability

The datasets presented in this study can be found in Appendix A.

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
