# Peer review of "Comparative Chloroplast Genomes Analysis Provided Adaptive Evolution Insights in *Medicago ruthenica"

_ijms, 2024, doi:10.3390/ijms25168689_

Round 1
Reviewer 1 Report
Comments and Suggestions for Authors
Medicago ruthenica is a frequent topic of the research. The work focuses on phylogenetic relationships and genetic diversity, especially attention to the divergence of chloroplast populations in different regions, which provides new chloroplast genomic evidence for the origin and evolution of Medicago ruthenica. The work focuses also on chloroplast genes involved in responding to abiotic stress and discussed their roles in mediating tolerance in Medicago ruthenica. The ,,abiotic stresses” are not precisely defined in the work. Moreover, the combination of undefined abiotic stresses and other genes is problematic. This problem need more explanation. And, why chloroplasts genome? How was it obtained? When, in which part of plant and why?
However, the work is generally very good prepared and compiled and can be published after minor corrections.
The abstract – the aims of the research should be highlighted.
The table 1 is divided. It should be entirely.
Line 253: the names – italics
Line 273: persica
Comments on the Quality of English LanguageThe minor editing only.
Author Response
Dear reviewer,
Thank you for your careful review work and positive comments, we have thoroughly revised our manuscript based your and other reviewers’ comments, thank you. The comments and responses were listed in point by point as flowing, and the revisions were highlighted in main text, you will find them. Thank you very much.
Q1: Medicago ruthenica is a frequent topic of the research. The work focuses on phylogenetic relationships and genetic diversity, especially attention to the divergence of chloroplast populations in different regions, which provides new chloroplast genomic evidence for the origin and evolution of Medicago ruthenica. The work focuses also on chloroplast genes involved in responding to abiotic stress and discussed their roles in mediating tolerance in Medicago ruthenica. The, abiotic stresses” are not precisely defined in the work. Moreover, the combination of undefined abiotic stresses and other genes is problematic. This problem need more explanation. And, why chloroplasts genome? How was it obtained? When, in which part of plant and why?
Answer 1: Thank you for your suggestion. We have revised the introduction section and provided a more detailed description. For example, the chloroplast genome contains important genetic information and are also the site of synthesis for many macromolecules, including amino acids and fatty acids. The chloroplast genome data we used were obtained from publicly available databases for Medicago ruthenica (PRJNA598783 and PRJNA692663), as stated in line 370 of the manuscript. We have made factual corrections in our manuscript. Thanks for your attention.
Q2: The abstract – the aims of the research should be highlighted.
Answer 2: Thank you for your suggestion. We have revised the abstract section and provided a more detailed description, for example, but also highlighted the importance of chloroplast transcriptome analysis in elucidating the model of chloroplast responses to abiotic stress. We have made factual corrections in our manuscript. Thanks for your attention.
Q3: The table 1 is divided. It should be entirely.
Answer 3: Thank you for your reminder. We have corrected table 1 and presented in the right form in the manuscript. We were extremely sorry for our carelessness. Thank you again for the reminder.
Q4: Line 253: the names – italics
Answer 4: Thank you very much for your valuable suggestion. We have revised the corresponding content in the manuscript. Thanks again for your suggestion.
Q5: Line 273: persica
Answer 5: Thank you very much for your valuable suggestion. We have corrected the names and, the previous citations were outdated, added more recent studies. The manuscript has been revised accordingly. Thank you for your suggestion.
Reviewer 2 Report
Comments and Suggestions for Authors
The manuscript deals with an interesting topic, but it is out of scope and themes published from the IJMS due to weak relevance and affiliation with in depth molecular analysis and mechanims of molecular transformation/transportation/translocation/metabolism. I recommend it to be transferred for peer review evaluation at the journals: Genes, or Plants MDPI. Other points of improvement prior to acceptance of publication in one of these journals, if submitted by authors, are the following:
1. Sections 2 and 4 have to interchange their positions. Besides, section 3. Discussion goes after the Results section.
2. The section of Literature Review is missing, thus, it has to be formulated from the beginning.
3. Both the lengthy blocks of narrative in sections Introduction, Discussion, have to be reorganized into 2-3 shorter subsections by also titling them, accordingly.
4. It is not clear if there are findings and remarks of generalized applicability, beyond the research focus on “total of 61 chloroplast genomes from eastern and western regions of China” , towards other genomes and other geographical areas which could be derived?, or come to conclusions of wider applicability? For this, the Discussion, Conclusions sections can be also informative, accordingly. All summary-written concluding remarks can be presented in the preceding sections, but underscoring in Conclusions only those constraints, challenges and future prospects of the research findings. For this 1-2 no numerical no cross-citing paragraphs are adequate.
5. Almost half of the citations are dating back to one decade ago, or even earlier. Therefore, authors are recommended to update the theoretical production of their study by adding more and fresher published studies, if applicable, thus, enriching their theoretical coverage in a more recent and pluralistic manner.
Author Response
Dear reviewer,
Thank you for your careful review work and positive comments, we have thoroughly revised our manuscript based your and other reviewers’ comments, thank you. The comments and responses were listed in point by point as flowing, and the revisions were highlighted in main text, you will find them. Thank you very much.
Q1: The manuscript deals with an interesting topic, but it is out of scope and themes published from the IJMS due to weak relevance and affiliation with in depth molecular analysis and mechanims of molecular transformation/transportation/translocation/metabolism. I recommend it to be transferred for peer review evaluation at the journals: Genes, or Plants MDPI. Other points of improvement prior to acceptance of publication in one of these journals, if submitted by authors, are the following:
- Sections 2 and 4 have to interchange their positions. Besides, section 3. Discussion goes after the Results section.
- The section of Literature Review is missing, thus, it has to be formulated from the beginning.
Answer 1: We appreciate your positive comments about our manuscript, as you see, there were many reports in journals Plants and Genes. However, as we searched in journal IJMS, there were also over 60 articles published with similar topics. Such as following:
1.Zhu, J.; Huang, Y.; Chai, W.; Xia, P. Decoding the Chloroplast Genome of Tetrastigma (Vitaceae): Variations and Phylogenetic Selection Insights. Int. J. Mol. Sci. 2024, 25, 8290. https://doi.org/10.3390/ijms25158290
- Luo, L.; Qu, Q.; Lin, H.; Chen, J.; Lin, Z.; Shao, E.; Lin, D. Exploring the Evolutionary History and Phylogenetic Relationships of Giant Reed (Arundo donax) through Comprehensive Analysis of Its Chloroplast Genome. Int. J. Mol. Sci.2024, 25, 7936. https://doi.org/10.3390/ijms25147936
- Wu, Y.; Zeng, M.-Y.; Wang, H.-X.; Lan, S.; Liu, Z.-J.; Zhang, S.; Li, M.-H.; Guan, Y. The Complete Chloroplast Genomes of Bulbophyllum (Orchidaceae) Species: Insight into Genome Structure Divergence and Phylogenetic Analysis. Int. J. Mol. Sci. 2024, 25, 2665. https://doi.org/10.3390/ijms25052665
We fell our manuscript is suitable for IJMS scope, please reconsider our manuscript for publishing on IJMS, we are appreciating and honor for your recommendation. Meanwhile, the manuscript was adjusted as you suggested, you will find them in revision. Thanks again.
Q2: Both the lengthy blocks of narrative in sections Introduction, Discussion, have to be reorganized into 2-3 shorter subsections by also titling them, accordingly.
Answer 2: Thank you for your reminder. We have reorganized the lengthy narrative into 2 to 3 shorter subsections and presented them in the correct form in the manuscript. We deeply apologize for our oversight. Thank you again for the reminder.
Q3: It is not clear if there are findings and remarks of generalized applicability, beyond the research focus on “total of 61 chloroplast genomes from eastern and western regions of China”, towards other genomes and other geographical areas which could be derived?, or come to conclusions of wider applicability? For this, the Discussion, Conclusions sections can be also informative, accordingly. All summary-written concluding remarks can be presented in the preceding sections, but underscoring in Conclusions only those constraints, challenges and future prospects of the research findings. For this 1-2 no numerical no cross-citing paragraphs are adequate.
Answer 3: Thank you for your suggestion. We have revised the Discussion and Conclusion sections to provide more detailed descriptions. For example, we have included information about the relationship between the wild varieties E1 and E4 and the cultivated varieties. In the Conclusion, we have also highlighted the limitations, challenges, and future prospects of the study. We have made factual corrections in the manuscript. Thank you for your attention.
Q4: Almost half of the citations are dating back to one decade ago, or even earlier. Therefore, authors are recommended to update the theoretical production of their study by adding more and fresher published studies, if applicable, thus, enriching their theoretical coverage in a more recent and pluralistic manner.
Answer 4: We sincerely apologize for our previous careless mistakes. Thank you for your reminder. We have included additional and updated published research in the manuscript and appreciate the reviewers for their careful reading. Thank you again.
Round 2
Reviewer 1 Report
Comments and Suggestions for Authors
The manuscript was corrected, but I have still concerns regards to the research object.
Authors have written in Abstract: Line 16 – cultivated? and wild?, then lines 19, 21 – clades, groups – which Medicago? Line 32 - Medicago varieties?
The research material should be specified in Materials and Methods. This is required.
Author Response
Dear reviewer,
Thank you for your suggestion about our manuscript, we have addressed all comments, and revised our manuscript, you will find them in manuscript. Meanwhile, the responses were also listed as follow:
Q1: The manuscript was corrected, but I have still concerns regards to the research object.
Answer 1: Thank you. We have revised our manuscript, and the object of our manuscript is distinguishing Medicago ruthenica germplasm, including cultivated varieties and wild germplasm. Meanwhile, we have characterized chloroplast genes with different profiles response to abiotic stress in alfalfa, therefore, we have investigated homologous genes in response to various stress from Medicago ruthenica. We have revised our manuscript for more clearly described, you will find them. Thank you very much.
Q2: Authors have written in Abstract: Line 16 – cultivated? and wild?, then lines 19, 21 – clades, groups – which Medicago? Line 32 - Medicago varieties?
Answer 2: Thank you for your attention, we have added Medicago ruthenica germplasm information in manuscript, including 16 cultivated varieties and 45 wild germplasm, and the corresponding information was listed in Table S1, you will find them. Thank you again.
Q3: The research material should be specified in Materials and Methods. This is required.
Answer 3: Thank you for your comments. We have revised our manuscript with more detailly methods description, including data accessing, chloroplast genome assembly, chloroplast genome comparison, etc. All revisions were highlighted, you will find them. Thank you very much.
Reviewer 2 Report
Comments and Suggestions for Authors
At this revised manuscript authors developed a satisfactory revision of their initial study, having systematically addressed the review comments. In this context this revised manuscript can be accepted for publication at the International Journal of Molecular Sciences as is.
Author Response
Dear reviewer,
Thank you for your valuable comments about our manuscript, and we are grateful for your recommendation about our manuscript. Thank you very much.
Round 3
Reviewer 1 Report
Comments and Suggestions for Authors
The manuscript was sufficiently corrected.
Line 189: Is it footnote?
Author Response
Q1: The manuscript was sufficiently corrected.
Line 189: Is it footnote?
Answer 1: Thank you for your positive comments about our manuscript. You are right, the line 189 is the footnote of Table 2, we have revised it. Thank you for your suggestion.